# Summary Measures of Health Inequality: A Review of Existing Measures and Their Application

**DOI:** 10.3390/ijerph19063697

**Published:** 2022-03-20

**Authors:** Anne Schlotheuber, Ahmad Reza Hosseinpoor

**Affiliations:** Department of Data and Analytics, World Health Organization, 1211 Geneva, Switzerland; schlotheuberan@who.int

**Keywords:** health equity, health inequality, summary measures, measurement, monitoring

## Abstract

Measuring and monitoring health inequalities is key to achieving health equity. While disaggregated data are commonly used to assess differences in health between different population subgroups, summary measures of health inequality also play a vital role in monitoring health inequalities. Building on disaggregated data, they quantify the level of inequality in a single number and are useful to compare inequality over time and across different health indicators, programmes and settings. We provide a comprehensive overview of existing summary measures of health inequality, including their definition, calculation, interpretation and application. The use of these measures is illustrated based on an example from the WHO’s Health Equity Monitor database using the WHO’s Health Equity Assessment Toolkit (HEAT) software. We discuss the strengths and limitations of different measures and provide guidance for selecting suitable summary measures for analysing health inequalities and communicating results. Summary measures of health inequality should form an integral part of health inequality monitoring to inform equity-oriented policies and programmes.

## 1. Introduction

Measuring and monitoring health inequalities is essential for achieving health equity—a core commitment of the World Health Organization (WHO) and a central goal of the 2030 Agenda for Sustainable Development [1,2]. Health inequality monitoring makes use of data about health inequalities (observable differences in health between different population subgroups) to inform policies and programmes that aim to tackle health inequities (health inequalities that are unfair, avoidable or remediable). In the context of the Sustainable Development Goals (SDGs), disaggregated data have gained particular visibility and recognition for inequality monitoring, with SDG target 17.18 explicitly calling for increased availability of high-quality, timely and reliable disaggregated data [3]. Indeed, disaggregated data are one important type of inequality data as they break down national averages and enable identifying patterns of inequality in a population and subgroups that are left behind. In addition to disaggregated data, summary measures of health inequality also play a vital role in monitoring health inequalities. Summary measures build on disaggregated data, quantifying the level of inequality in a population in a single number, thereby allowing for an easy comparison of health inequality over time and across different settings and indicators.

A number of earlier works from the 1990s and 2000s have described existing summary measures of health inequality [4,5,6,7,8,9,10]. One of the key debates at the turn of the century was around measuring total versus social inequality in health [11,12,13]. Measures of total inequality, such as the Gini coefficient, only consider the distribution of a health indicator in a population, while measures of social inequality assess how a health indicator varies according to different demographic, socioeconomic or geographic characteristics. In the context of health inequality monitoring, the term “summary measures of health inequality” usually refers to measures of social inequality as this is the main focus of interest. A large number of summary measures exists, each with different characteristics, that can lead to different conclusions about the magnitude and direction of inequality [5,9]. It is therefore important to consider several methodological issues when calculating summary measures and choosing a suite of appropriate measures for reporting results [10,14].

The aim of this paper is to provide an updated and systematic overview of existing summary measures of health inequality, all of which are available via the WHO’s Health Equity Assessment Toolkit (HEAT) [15]. The software application was developed to facilitate the assessment of health inequalities in countries and calculates a range of summary measures based on disaggregated data. Following a discussion on essential methodological considerations for calculating summary measures, we provide a detailed description of the measures calculated in HEAT, including their definition, calculation, interpretation and application. The use of these measures is illustrated based on an example from the WHO’s Health Equity Monitor database. Following the presentation of results, we discuss the strengths and limitations of different measures and provide guidance on selecting suitable measures for the analysis and effectively communicating results.

## 2. Essential Considerations for Calculating Summary Measures of Health

There are a number of essential considerations that need to be considered when calculating summary measures of health inequality [14,16,17]. Firstly, not all measures can be calculated for all types of data, so the characteristics of the underlying data about health indicators and inequality dimensions need to be taken into account. Secondly, different measures assess different aspects of inequality, and it is therefore important to consider the inherent properties of each summary measure and the intended purpose of the analysis. It is worth noting at this point that some summary measures can be calculated based on both individual-level or group-level data. However, for the purpose of this review, we focus on the calculation of summary measures based on group-level data, or disaggregated data, only. The following section briefly describes characteristics that are relevant for the calculation of summary measures.

### 2.1. Health Indicators

Health indicators may measure various aspects of health, including inputs and processes, outputs, outcomes and impact, as outlined in the WHO’s monitoring, evaluation and review framework [18,19]. The WHO Global Reference List of 100 core health indicators presents a set of standard health indicators that are prioritised for global and national health monitoring [20]. Examples include health facility density (input), access to medicines (output), antenatal care coverage (outcome) and under-five mortality rate (impact). Each indicator has a defined measurement unit (such as number, rate, proportion or percentage) and an optimal level that is to be achieved or maintained through public health action. For two types of health indicators, the optimal level is clearly defined: for favourable indicators, the aim is to attain a maximum level, such as complete coverage of antenatal care or the highest possible life expectancy; while for adverse indicators, the goal is to realise a minimum level, such as zero stunting prevalence or zero mortality rate in children under five. There are also health indicators that do not fall into either of these two categories, such as fertility rates, caesarean section rates or hospitalisation rates. For these indicators, the optimum is neither the maximum nor the minimum, but very much depends on the setting and context.

### 2.2. Inequality Dimensions

Dimensions of inequality refer to demographic, socioeconomic or geographic characteristics, based on which populations can be categorised into different subgroups [21,22]. Inequality dimensions that are commonly used for global health inequality monitoring due to good availability of comparable data for a large number of countries are: age, economic status, education, place of residence and sex [23]. These dimensions are also recommended as a basis for data disaggregation by the SDGs, as well as several other dimensions that may be relevant, depending on the setting and context, such as disability, ethnicity, language, migratory status, race and subnational region [3]. At the most basic level, these dimensions can be divided into dimensions that compare the situation in two population subgroups (such as place of residence, i.e., urban and rural areas), and dimensions that look at the situation in more than two population subgroups (such as wealth quintiles and subnational regions). In the case of dimensions with more than two subgroups, it is possible to further differentiate between dimensions with ordered subgroups and non-ordered subgroups. Ordered dimensions have subgroups with an inherent ordering, such as wealth quintiles, that can be ranked from poorest to richest. Non-ordered dimensions, by contrast, have subgroups that cannot be logically ranked, such as subnational regions.

### 2.3. Summary Measures

Generally, summary measures of health inequality assess either absolute or relative inequality. While absolute measures indicate the magnitude of inequality between population subgroups and normally retain the unit of the health indicator, relative measures show proportional inequality between subgroups and have no unit. Furthermore, summary measures can be divided into simple and complex measures of inequality. Simple measures make pairwise comparisons between two population subgroups, such as the poorest and richest wealth quintile. Complex measures, on the other hand, consider the situation in all population subgroups (including, for example, the three middle wealth quintiles) and they may also account for the population size of each subgroup. Many complex measures are weighted measures, i.e., weighting population subgroups according to their population size, whereas simple measures are unweighted measures, i.e., weighting all subgroups equally regardless of their size. There are two main types of complex measures: ordered measures, that can be calculated for ordered dimensions (such as economic status) and non-ordered measures, that can be calculated for non-ordered dimensions (such as subnational region). An important consideration for non-ordered measures is the selection of a reference point against which the other subgroups are compared. The reference point may be a selected reference subgroup (e.g., the best-performing subgroup), a reference value (e.g., the national, provincial or district average for analyses at the national, provincial or district level), or a defined target. Impact measures are another type of complex measures that can be calculated for both ordered and non-ordered dimensions. These types of measures assess the impact of addressing inequality and estimate the possible improvement in setting average of a health indicator that could be achieved if inequality was eliminated.

## 3. Summary Measures of Health Inequality

Figure 1 provides an overview of the summary measures of health inequality calculated in HEAT. The following section provides a detailed description of these measures, including their definition, calculation and interpretation. A summary of these characteristics is provided in Appendix A (Table A1). The software also calculates 95% confidence intervals for these measures, either based on a formula or based on a simulation (see Table A2).

### 3.1. Simple Measures

Simple measures compare the situation in two population subgroups. The selection of the two subgroups depends on the characteristics of the inequality dimension and the purpose of the analysis. For dimensions with two subgroups, such as place of residence (urban/rural areas), the selection is straightforward. For ordered dimensions, typically, the most-advantaged and most-disadvantaged subgroups are compared (e.g., the poorest and richest wealth quintile). For non-ordered dimensions, the subgroups with the lowest and highest indicator value can be used (e.g., the subnational region with the lowest and highest value). Alternatively, it is possible to compare the situation between a defined reference subgroup (e.g., the capital city) or a defined reference value (such as the setting average or a target value) and another population subgroup. In some cases, it may also be appropriate to compare the situation between defined top and bottom percentiles (e.g., the 5th and 95th percentile of districts).


**Difference**


Definition: The difference (D) is an absolute measure of inequality that shows the difference in health indicator between two population subgroups.

Calculation: D can be calculated as:

$$D=y_{1}-y_{2}$$
 where 
y1
 and 
$y_{2}$
 indicate the estimates for subgroups 1 and 2. As described above, the selection of the two subgroups depends on the characteristics of the inequality dimension and the purpose of the analysis. In addition, the direction of the calculation may depend on the indicator type (favourable vs. adverse). Please refer to Appendix A for details about the calculation of D in HEAT (Table A3).

Interpretation: If there is no inequality, D assumes the value of zero. Greater absolute values indicate higher levels of inequality.


**Ratio**


Definition: The ratio (R) is a relative measure of inequality that shows the ratio of two population subgroups.

Calculation: R can be calculated as:

$$R=y_{1}/y_{2}$$
 where 
y1
 and 
$y_{2}$
 indicate the estimates for subgroups 1 and 2. As described above, the selection of the two subgroups depends on the characteristics of the inequality dimension and the purpose of the analysis. In addition, the direction of the calculation may depend on the indicator type (favourable vs. adverse). Please refer to Appendix A for details about the calculation of R in HEAT (Table A4).

Interpretation: If there is no inequality, R assumes the value of one. R assumes only positive values. The further the value of R from one, the higher the level of inequality. R is a multiplicative measure and has to be displayed on a logarithmic scale (values larger than one are equivalent in magnitude to their reciprocal values smaller than one, e.g., a value of 2 is equivalent in magnitude to a value of 0.5).

### 3.2. Complex Measures

Complex measures are calculated for inequality dimensions with more than two population subgroups. They consider the situation in all population subgroups and they may also account for the population share of each subgroup. We can differentiate between three types of complex measures as described below: ordered, non-ordered and impact measures.

#### 3.2.1. Ordered Measures

Ordered measures can be calculated for inequality dimension with subgroups that have a natural ordering, such as wealth quintiles. They can be grouped into two categories: disproportionality measures, which express inequality as a function of shares of the health indicator compared to shares of the population, and regression-based measures, which make use of an appropriate regression model to estimate the association between the rank of subgroups and the health indicator [10,24].


*Disproportionality Measures*

**Absolute Concentration Index**


Definition: The absolute concentration index (ACI) is an absolute measure of inequality that shows the gradient across population subgroups. It indicates the extent to which an indicator is concentrated among disadvantaged or advantaged subgroups, on an absolute scale. Subgroups are weighted according to their population share.

Calculation: The calculation of ACI is based on a ranking of the whole population from the most-disadvantaged subgroup (at rank 0) to the most-advantaged subgroup (at rank 1), which is inferred from the ranking and size of the subgroups. The relative rank of each subgroup is calculated as: 
$X_{j}=\underset{j}{\sum }p_{j}-0.5p_{j}$
. Based on this ranking, ACI can be calculated as:
$$ACI=\sum_{j}p_{j}\left(2X_{j}-1\right)y_{j}$$

where 
yj
 indicates the estimate for subgroup *j*, 
pj
 the population share of subgroup *j* and 
$X_{j}$
 the relative rank of subgroup *j*.

Interpretation: If there is no inequality, ACI assumes the value of zero. Positive values indicate a concentration of the indicator among the advantaged, while negative values indicate a concentration of the indicator among the disadvantaged. The larger the absolute value of ACI, the higher the level of inequality.


**Relative Concentration Index**


Definition: The relative concentration index (RCI) is a relative measure of inequality that shows the gradient across population subgroups, on a relative scale. It indicates the extent to which an indicator is concentrated among disadvantaged or advantaged subgroups. Subgroups are weighted according to their population share.

Calculation: RCI is calculated by dividing the absolute concentration index (ACI) by the setting average 
μ
. RCI may be more easily interpreted when multiplied by 100:
$$RCI=\frac{ACI}{\mu }*100$$


Interpretation: RCI is bounded between −1 and +1 (or between −100 and +100, when multiplied by 100) and assumes the value of zero if there is no inequality. Positive values indicate a concentration of the indicator among the advantaged, while negative values indicate a concentration of the indicator among the disadvantaged. The greater the absolute value of RCI, the higher the level of inequality.


*Regression-Based Measures*

**Slope Index of Inequality**


Definition: The slope index of inequality (SII) is an absolute measure of inequality that represents the difference in estimated indicator values between the most-advantaged and most-disadvantaged subgroup, while taking into consideration the situation in all other subgroups—using an appropriate regression model. Subgroups are weighted according to their population share.

Calculation: To calculate SII, a weighted sample of the whole population is ranked from the most-disadvantaged subgroup (at rank 0) to the most-advantaged subgroup (at rank 1). This ranking is weighted, accounting for the proportional distribution of the population within each subgroup. The population of each subgroup is then considered in terms of its range in the cumulative population distribution, and the midpoint of this range. The indicator of interest is then regressed against this midpoint value using an appropriate regression model (e.g., a generalised linear model with logit link), and the predicted values of the indicator are calculated for the two extremes (rank 1 and rank 0). The difference between the estimated values at rank 1 (
$v_{1}$
) and rank 0 (
$v_{0}$
) (covering the entire distribution) generates the SII value:
$$SII=v_{1}-v_{0}$$


Interpretation: If there is no inequality, SII assumes the value of zero. Greater absolute values indicate higher levels of inequality. Positive values indicate a concentration of the indicator among the advantaged and negative values indicate a concentration of the indicator among the disadvantaged.


**Relative Index of Inequality**


Definition: The relative index of inequality (RII) is a relative measure of inequality that represents the ratio of estimated indicator values of the most-advantaged to the most-disadvantaged subgroup, while taking into account the situation in all other subgroups—using an appropriate regression model. Subgroups are weighted according to their population share.

Calculation: To calculate RII, a weighted sample of the whole population is ranked from the most-disadvantaged subgroup (at rank 0) to the most-advantaged subgroup (at rank 1). This ranking is weighted, accounting for the proportional distribution of the population within each subgroup. The population of each subgroup is then considered in terms of its range in the cumulative population distribution, and the midpoint of this range. The indicator of interest is then regressed against this midpoint value using an appropriate regression model (e.g., a generalised linear model with logit link), and the predicted values of the indicator are calculated for the two extremes (rank 1 and rank 0). The ratio of the estimated values at rank 1 (
$v_{1}$
) to rank 0 (
$v_{0}$
) (covering the entire distribution) generates the RII value:
$$RII=v_{1}/v_{0}$$


Interpretation: If there is no inequality, RII assumes the value of one. RII assumes only positive values. The further the value of RII from one, the higher the level of inequality. Values larger than one indicate a concentration of the indicator among the advantaged and values smaller than one indicate a concentration of the indicator among the disadvantaged. RII is a multiplicative measure and has to be displayed on a logarithmic scale (values larger than one are equivalent in magnitude to their reciprocal values smaller than one, e.g., a value of 2 is equivalent in magnitude to a value of 0.5).

#### 3.2.2. Non-Ordered Measures

Non-ordered measures can be calculated for dimensions with subgroups that do not have a natural ordering, such as subnational regions. There are three main groups of non-ordered measures: variance measures, which are based on the variance that summarises the squared deviations from the setting average; mean difference measures, which show the mean difference from a reference point, such as the setting average or the best-performing subgroup, i.e., the subgroup with the highest value in the case of favourable health indicators and the subgroup with the lowest value in the case of adverse health indicators; and disproportionality measures, which express inequality as a function of shares of the health indicator compared to shares of the population [10,25,26]. Variance measures give more weight to the extremes; mean difference measures weigh all differences equally.


*Variance Measures*

**Between-Group Variance**


Definition: The between-group variance (BGV) is an absolute measure of inequality that considers all population subgroups. Subgroups are weighted according to their population share.

Calculation: BGV is calculated as the weighted average of squared differences between the subgroup estimates 
$y_{j}$
 and the setting average 
μ
. Squared differences are weighted by each subgroup’s population share 
pj
:
$$BGV=\underset{j}{\sum }p_{j}{\left(y_{j}-\mu \right)}^{2}$$


Interpretation: BGV obtains only positive values with larger values indicating higher levels of inequality. BGV is zero if there is no inequality. BGV is more sensitive to outlier estimates as it gives more weight to the estimates that are further from the setting average. It is reported as the squared unit of the health indicator.


**Between-Group Standard Deviation**


Definition: The between-group standard deviation (BGSD) is an absolute measure of inequality that considers all population subgroups. Subgroups are weighted according to their population share.

Calculation: BGSD is calculated as the square root of the weighted average of squared differences between the subgroup estimates 
$y_{j}$
 and the setting average 
μ
. Squared differences are weighted by each subgroup’s population share 
pj
:
$$BGSD=\sqrt{\underset{j}{\sum }p_{j}{\left(y_{j}-\mu \right)}^{2}}$$


Interpretation: BGSD assumes only positive values, with larger values indicating higher levels of inequality. BGSD is zero if there is no inequality. BGSD is more sensitive to outlier estimates as it gives more weight to the estimates that are further from the setting average. It has the same unit as the health indicator.


**Coefficient of Variation**


Definition: The coefficient of variation (COV) is a relative measure of inequality that considers all population subgroups. Subgroups are weighted according to their population share.

Calculation: COV is calculated by dividing the between-group standard deviation (BGSD) by the setting average 
μ
 and multiplying the fraction by 100:
$$COV=\frac{BGSD}{\mu }*100$$


Interpretation: COV assumes only positive values, with larger values indicating higher levels of inequality. COV is zero if there is no inequality.


*Mean Difference Measures*

**Mean Difference from Mean**


Definition: The mean difference from mean (MDM) is an absolute measure of inequality that shows the mean difference between each subgroup and the setting average. MDM can be calculated as an unweighted or weighted measure. For the unweighted version, all subgroups are weighted equally. For the weighted version, subgroups are weighted according to their population share.

Calculation: The unweighted version (MDMU) is calculated as the average of absolute differences between the subgroup estimates 
$y_{j}$
 and the setting average 
μ
, divided by the number of subgroups 
n
:
$$MDMU=\frac{1}{n}*\underset{j}{\sum }\left|y_{j}-\mu \right|$$

The weighted version (MDMW) is calculated as the weighted average of absolute differences between the subgroup estimates 
$y_{j}$
 and the setting average 
μ
. Absolute differences are weighted by each subgroup’s population share 
$p_{j}$
:
$$MDMW=\underset{j}{\sum }p_{j}\left|y_{j}-\mu \right|$$


Interpretation: MDM assumes only positive values, with larger values indicating higher levels of inequality. MDM is zero if there is no inequality.


**Mean Difference from Best-Performing Subgroup**


Definition: The mean difference from best-performing subgroup (MDB) is an absolute measure of inequality that shows the mean difference between each population subgroup and a reference subgroup. MDB can be calculated as an unweighted or weighted measure. For the unweighted version, all subgroups are weighted equally. For the weighted version, subgroups are weighted according to their population share.

Calculation: The unweighted version (MDBU) is calculated as the average of absolute differences between the subgroup estimates 
$y_{j}$
 and the estimate for the reference subgroup 
$y_{ref}$
, divided by the number of subgroups 
n
:
$$MDBU=\frac{1}{n}*\underset{j}{\sum }\left|y_{j}-y_{ref}\right|$$
The weighted version (MDBW) is calculated as the weighted average of absolute differences between the subgroup estimates 
$y_{j}$
 and the estimate for the reference subgroup 
$y_{ref}$
. Absolute differences are weighted by each subgroup’s population share 
pj
:
$$MDBW=\underset{j}{\sum }p_{j}\left|y_{j}-y_{ref}\right|$$


$y_{ref}$
 refers to the subgroup with the highest estimate in the case of favourable indicators and to the subgroup with the lowest estimate in the case of adverse indicators.

Interpretation: MDB assumes only positive values, with larger values indicating higher levels of inequality. MDB is zero if there is no inequality.


**Index of Disparity**


Definition: The index of disparity (IDIS) is a relative measure of inequality that shows the average difference between each population subgroup and the setting average, in relative terms. IDIS can be calculated as an unweighted or weighted measure. For the unweighted version, all subgroups are weighted equally. For the weighted version, subgroups are weighted according to their population share.

Calculation: The unweighted version (IDISU) is calculated as the average of absolute differences between the subgroup estimates 
$y_{j}$
 and the setting average 
μ
, divided by the number of subgroups 
n
 and the setting average 
μ
, and multiplied by 100:
$$IDISU=\frac{\frac{1}{n}*\underset{j}{\sum }\left|y_{j}-\mu \right|}{\mu }*100$$
The weighted version (IDISW) is calculated as the weighted average of absolute differences between the subgroup estimates 
$y_{j}$
 and the setting average 
μ
, divided by the setting average 
μ
, and multiplied by 100. Absolute differences are weighted by each subgroup’s population share 
$p_{j}$
:
$$IDISW=\frac{\underset{j}{\sum }p_{j}\left|y_{j}-\mu \right|}{\mu }*100$$


Interpretation: IDIS assumes only positive values, with larger values indicating higher levels of inequality. IDIS is zero if there is no inequality.


*Disproportionality Measures*

**Theil Index**


Definition: The Theil index (TI) is a relative measure of inequality that considers all population subgroups. Subgroups are weighted according to their population share.

Calculation: TI is calculated as the sum of products of the natural logarithm of the share of the indicator of each subgroup (
$ln\frac{y_{j}}{\mu }$
), the share of the indicator of each subgroup (
$\frac{y_{j}}{\mu }$
) and the population share of each subgroup (
$p_{j}$
). TI may be more easily interpreted when multiplied by 1000:
$$TI=\underset{j}{\sum }p_{j}\frac{y_{j}}{\mu }\ln\frac{y_{j}}{\mu }*1000$$

where 
yj
 indicates the estimate for subgroup *j*, 
pj
 the population share of subgroup *j* and 
μ
 the setting average.

Interpretation: If there is no inequality, TI obtains the value of zero. Greater absolute values indicate higher levels of inequality. TI is more sensitive to differences further from the setting average (by the use of the logarithm).


**Mean Log Deviation**


Definition: The mean log deviation (MLD) is a relative measure of inequality that considers all population subgroups. Subgroups are weighted according to their population share.

Calculation: MLD is calculated as the sum of products between the negative natural logarithm of the share of the indicator of each subgroup (
$-\ln\left(\frac{y_{j}}{\mu }\right)$
) and the population share of each subgroup (
$p_{j}$
). MLD may be more easily readable when multiplied by 1000:
$$\mathrm{MLD}=\sum_{j}p_{j}\left(-\ln\left(\frac{y_{j}}{\mu }\right)\right)*1000$$

where 
yj
 indicates the estimate for subgroup *j*, 
pj
 the population share of subgroup *j* and 
μ
 the setting average.

Interpretation: If there is no inequality, MLD assumes the value of zero. Greater absolute values indicate higher levels of inequality. MLD is more sensitive to differences further from the setting average (by the use of the logarithm).

#### 3.2.3. Impact Measures

Impact measures estimate the impact of addressing inequalities in a health indicator. They quantify the possible improvement in setting average of a health indicator that could be achieved if the entire population had the same level of the health indicator as a reference subgroup. Which subgroup is selected as the reference subgroup depends on the type of health indicator and inequality dimension. For ordered dimensions, the most-advantaged subgroup is used (e.g., the richest wealth quintile), regardless of the indicator type. For Non-ordered dimensions, such as subnational regions, usually the best-performing subgroup is used, i.e., the subgroup with the highest value in the case of favourable indicators and the subgroup with the lowest value in the case of adverse indicators. Alternatively, a defined reference subgroup can be used (e.g., the capital city).


**Population Attributable Risk**


Definition: The population attributable risk (PAR) is an absolute measure of inequality that shows the potential improvement in setting the average of a health indicator, in absolute terms, that could be achieved if all population subgroups had the same level of the indicator as a reference group.

Calculation: PAR is calculated as the difference between the estimate for the reference subgroup 
$y_{ref}$
 and the setting average *μ*:
$$PAR=y_{ref}-\mu$$
Given the setting average is the weighted average of all subgroup estimates (with subgroups being weighted by their population share), PAR considers all population subgroups. As described above, the selection of the reference subgroup depends on the characteristics of the inequality dimension and the indicator type. Please refer to Appendix A for details about the calculation of PAR in HEAT (Table A4).

Interpretation: PAR assumes positive values for favourable indicators and negative values for adverse indicators. The larger the absolute value of PAR, the higher the level of inequality. PAR is zero if no further improvement can be achieved, i.e., if all subgroups have reached the same level of the indicator as the reference subgroup or surpassed that level.


**Population Attributable Fraction**


Definition: The population attributable fraction (PAF) is a relative measure of inequality that shows the potential improvement in setting the average of a health indicator, in relative terms, that could be achieved if all population subgroups had the same level of the indicator as a reference group.

Calculation: PAF is calculated by dividing the population attributable risk (PAR) by the setting average 
μ
 and multiplying the fraction by 100:
$$PAF=\frac{PAR}{\mu }*100$$


Interpretation: PAF assumes positive values for favourable indicators and negative values for adverse indicators. The larger the absolute value of PAF, the larger the level of inequality. PAF is zero if no further improvement can be achieved, i.e., if all subgroups have reached the same level of the indicator as the reference subgroup or surpassed that level.

## 4. Application: WHO Health Equity Monitor Database

### 4.1. Materials and Methods

The summary measures described above were calculated based on disaggregated data from the WHO Health Equity Monitor database (2021 update) [23]. The database currently contains data for more than 30 reproductive, maternal, newborn and child health (RMNCH) indicators disaggregated by up to six inequality dimensions. Data are based on the re-analysis of publicly available microdata from more than 450 Demographic and Health Surveys (DHS), Multiple Indicator Cluster Surveys (MICS) and Reproductive Health Surveys (RHS) conducted in 115 countries in 1991–2019.

For the purpose of this paper, we used data from three rounds of DHS from Indonesia (1997, 2007 and 2017) to assess the latest situation of inequality and the change in inequality over time. In addition, we compared the situation in Indonesia with the situation in 15 other countries from the WHO South-East Asia and Western Pacific regions, using data from the latest available DHS and MICS from 2015–2019. We selected three RMNCH indicators, including births attended by skilled health personnel; demand for family planning satisfied (use of modern and traditional methods); and measles immunisation coverage among one-year-olds. All indicators were disaggregated by economic status and subnational region. Detailed information about the data sources, health indicators and inequality dimensions can be found in the indicator compendium of the Health Equity Monitor database [23].

Summary measures of health inequality and corresponding 95% confidence intervals were calculated using the HEAT software [15].

### 4.2. Results

Figure 2 shows the change over time in economic-related inequality in three reproductive, maternal and child health indicators in Indonesia. Overall, coverage increased and economic-related inequality decreased between 1997 and 2017 for all three indicators. The greatest improvements could be observed for births attended by skilled health personnel, with overall coverage almost doubling (national average of 51.0% in 1997 compared with 91.6% in 2017) and absolute economic-related inequality more than halving. For example, the slope index of inequality indicates a reduction in estimated difference between the richest and poorest quintile from 72.9 percentage points in 1997 to 31.6 percentage points in 2017 (Figure 3). Nevertheless, births attended by skilled health personnel remained the indicator with the largest economic-related inequality in 2017. There were only moderate inequalities in measles immunisation coverage among one-year-olds (SII of 15.5 percentage points) and no inequalities in demand for family planning satisfied (SII of −1.0 percentage points). The detailed estimates of disaggregated data and summary measures are available in Appendix A (Table A6 and Table A7).

The latest available data for these three health indicators show that coverage in Indonesia also varied largely by subnational region (Figure 4). In 2017, subnational–regional inequalities were largest for measles immunisation coverage among one-year-olds, with coverage being 40.0 percentage points or 1.8 times higher in the region with the highest coverage (91.4% in Bali) compared to the region with the lowest coverage (53.5% in Aceh). The weighted mean difference from mean indicates that, on average, measles immunisation coverage in subnational regions differed by 7.6 percentage points from the national average of 78.8%. Moderate subnational regional inequalities could be observed for births attended by skilled health personnel (MDMW of 5.3 percentage points) and low inequalities for demand for family planning satisfied (MDMW of 2.9 percentage points). The detailed estimates of disaggregated data and summary measures are available in Appendix A (Table A7 and Table A8).

In comparison with 15 other countries from the WHO South-East Asia and Western Pacific regions, Indonesia performed well overall, both in terms of national average (91.6%) and absolute economic-related inequality (difference of 23.6 percentage points between the richest and poorest quintile) in births attended by skilled health personnel (Figure 5). Countries fall into three clusters. Countries with similarly high coverage (national average >80.0%) and rather high inequality (difference between 20.0 and 30.0 percentage points), such as Indonesia, include Cambodia, India, the Philippines and Vietnam. There were some countries that had a better situation, such as Kiribati, Maldives, Mongolia, Thailand and Tonga, with similarly high coverage (national average >90.0%), but small inequality (absolute difference <5 percentage points). On the other hand, there were also several countries that had a worse situation, with lower coverage (national average <70.0%) and much larger inequality (difference > 50 percentage points).

Please refer to Appendix A for an interpretation of these results (Text A1). The 2017 *State of health inequality: Indonesia* report provides a comprehensive assessment of the state of health inequality in Indonesia [27].

## 5. Discussion

We have provided a systematic overview of existing summary measures of health inequality, including their definition, calculation and interpretation. The application of these measures based on an example from the WHO’s Health Equity Monitor database using the WHO’s Health Equity Assessment Toolkit demonstrates their usefulness for monitoring health inequalities in countries.

The different summary measures presented in this paper have different strengths and limitations that are important to keep in mind when monitoring health inequalities. Simple measures can be calculated for all types of inequality dimensions and are easy to interpret. However, for dimensions with more than two population subgroups, they ignore the situation in the other subgroups, and they do not account for the population size of each subgroup (simple measures are unweighted measures, i.e., weighting all population subgroups equally, regardless of their size). Complex measures overcome these two limitations. They consider all population subgroups and they may also account for the population share of each subgroup (many complex measures are weighted measures, i.e., weighting population subgroups according to their population size). However, complex measures are inherently more challenging to calculate and interpret. Whether to use simple or complex, unweighted or weighted measures, depends on the available data and purpose of the analysis. Similarly, the selection of a reference point for certain summary measures (such as the most-advantaged or best-performing subgroup, or the setting average or a target value), is a normative decision that needs to be considered in the context of the research question at hand. Finally, while summary measures of health inequality are useful to quantify the magnitude and direction of inequality, they do not explain why inequalities exist. In order to identify the root causes of inequality, further in-depth research is required that makes use of additional quantitative and qualitative data and more sophisticated analysis methods.

Software applications such as HEAT make it easy for users to calculate summary measures of health inequality, including complex measures [15]. The challenge here lies in selecting suitable measures for the analysis and effectively communicating the results. Figure 6 provides a decision tree for selecting appropriate measures based on the characteristics of the underlying data about health indicators and inequality dimensions. In addition, it is important to keep several things in mind when reporting results. Firstly, the needs and technical expertise of the target audience need to be considered. Simple measures are easy to understand and suitable for non-technical users, whereas complex measures provide a more nuanced picture and can be used for technical reports. When both simple and complex measures lead to the same conclusions, it is generally advisable to report simple measures. Secondly, both absolute and relative measures should be reported as they measure different aspects of inequality that complement each other. While absolute measures usually retain the unit of the health indicator and quantify the magnitude of inequality, relative measures have no unit and are useful to compare the situation across health indicators with different units. Multiplicative relative measures, such as the ratio and relative index of inequality, must be displayed on a logarithmic scale in order to adequately present the magnitude of inequality [28]. Thirdly, summary measures and disaggregated data should be presented alongside each other. Summary measures are useful to compare the situation across different settings, indicators and time points as they quantify the level of inequality in a single number. Disaggregated data can provide further detail, allowing users to identify patterns of inequality in a population and subgroups that are left behind. It may also be helpful to provide additional information, such as the population share of subgroups and confidence intervals as well as sample sizes (in the case of survey-based data). Finally, setting averages should be reported together with inequality data in order to provide a picture of the overall situation.

## 6. Conclusions

Summary measures of health inequality quantify the level of inequality in a single number and facilitate the assessment of inequalities over time and across different indicators and settings. Software applications, such as the WHO’s Health Equity Assessment Toolkit, make it easy for users to calculate, interpret and communicate summary measures. Their use should be complementary with disaggregated data, and form an integral part of health inequality monitoring in order to design health policies and programmes that are equity oriented.

## Figures and Tables

**Figure 1 ijerph-19-03697-f001:**
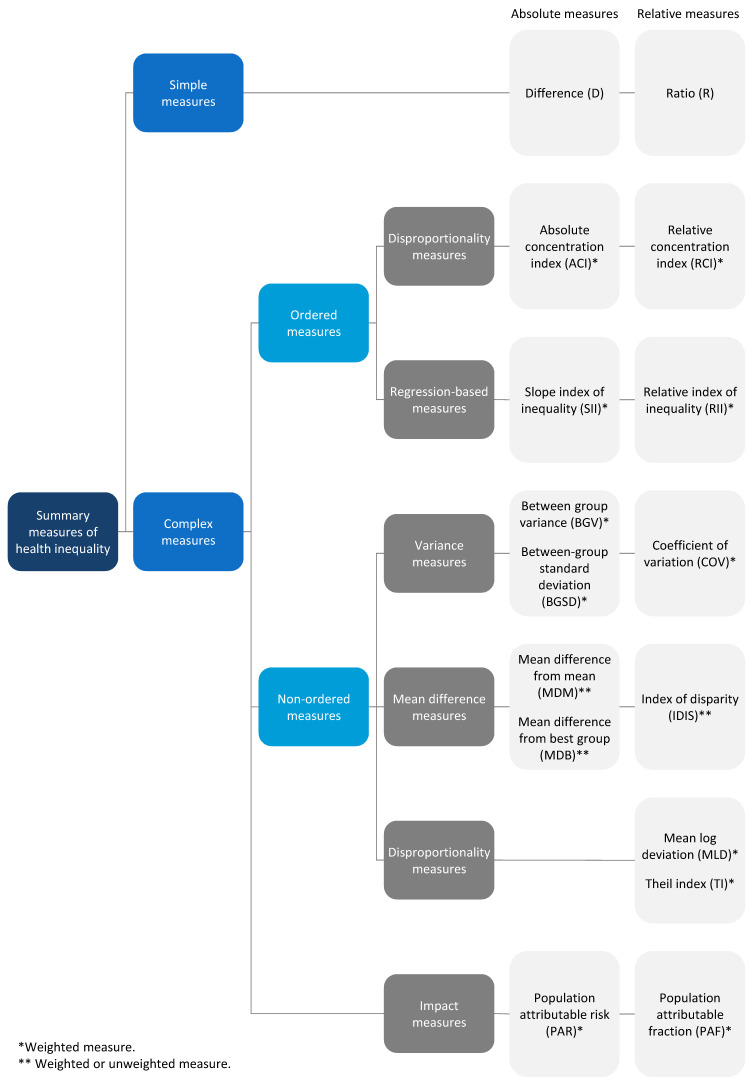
Overview of summary measures of health inequality.

**Figure 2 ijerph-19-03697-f002:**
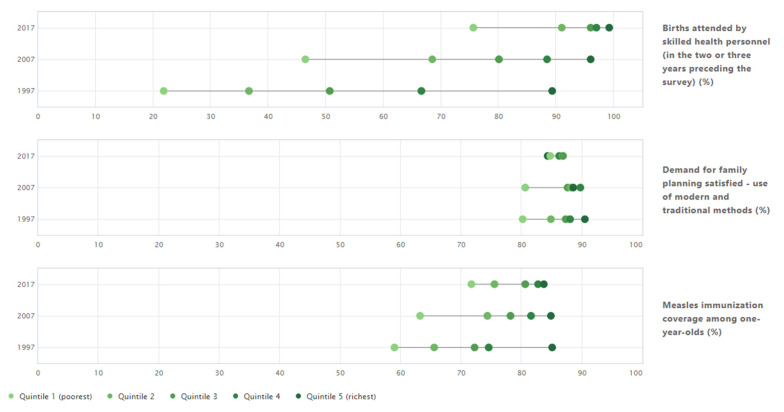
Three reproductive, maternal and child health indicators disaggregated by economic status: Indonesia (DHS: 1997, 2007 and 2017). Note: Circles indicate wealth quintiles. The horizontal lines indicate the difference between the most extreme wealth quintile values. Data source: WHO Health Equity Monitor database (2021 update).

**Figure 3 ijerph-19-03697-f003:**
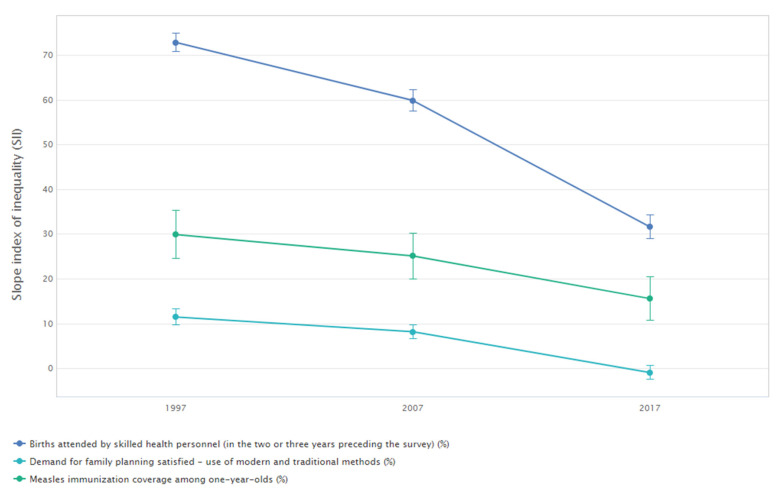
Absolute economic-related inequality in three reproductive, maternal and child health indicators: Indonesia (DHS: 1997, 2007 and 2017). Note: Vertical lines around point estimates indicate 95% confidence intervals. Data source: Summary measures calculated using the WHO Health Equity Assessment Toolkit (HEAT), based on disaggregated data from the WHO Health Equity Monitor database (2021 update).

**Figure 4 ijerph-19-03697-f004:**
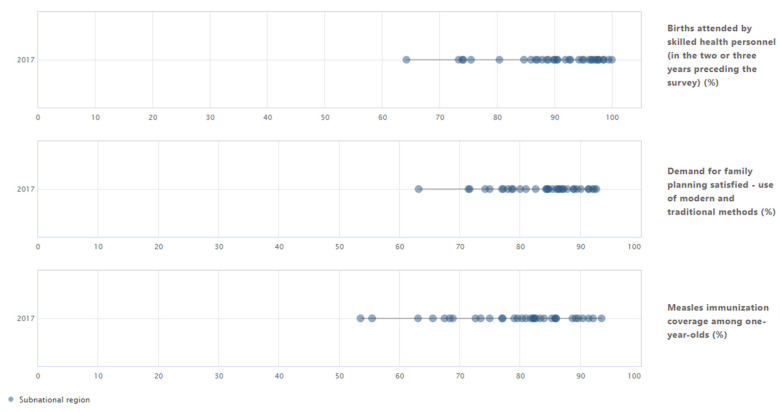
Three reproductive, maternal and child health indicators disaggregated by subnational region: Indonesia (DHS 2017). Note: Circles indicate subnational regions. The horizontal lines indicate the difference between the most extreme subnational regional values. Data source: WHO Health Equity Monitor database (2021 update).

**Figure 5 ijerph-19-03697-f005:**
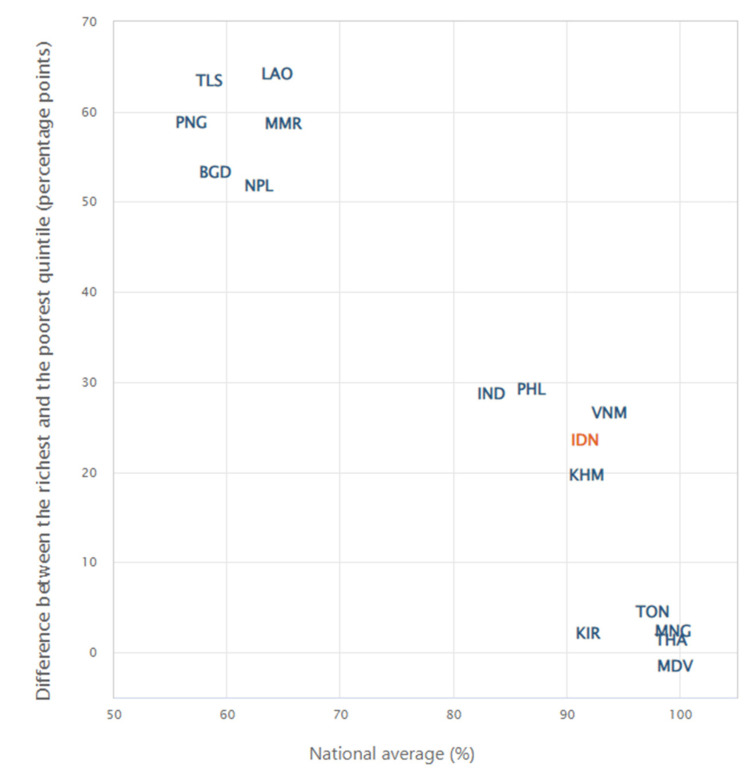
National average and absolute economic-related inequality in births attended by skilled health personnel in 16 countries from the WHO South-East Asia and Western Pacific regions (DHS and MICS, 2015–2019). Note: Countries shown as ISO 3 country codes. Data source: Summary measures calculated using the WHO Health Equity Assessment Toolkit (HEAT), based on disaggregated data from the WHO Health Equity Monitor database (2021 update).

**Figure 6 ijerph-19-03697-f006:**
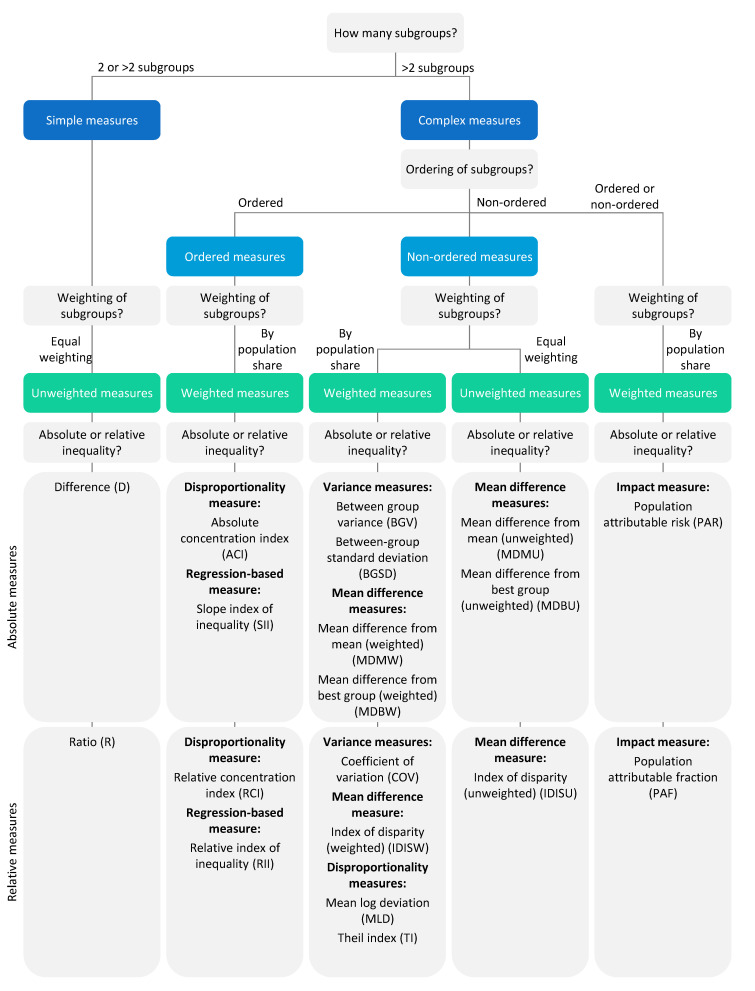
Decision tree for selecting appropriate summary measures of health inequality.

## Data Availability

The data presented in this study are openly available via the WHO Health Equity Monitor database (https://www.who.int/data/gho/health-equity/health-equity-monitor-database accessed on 10 January 2021).

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
