# Peer review of "Summary Measures of Health Inequality: A Review of Existing Measures and Their Application"

_ijerph, 2022, doi:10.3390/ijerph19063697_

Round 1
Reviewer 1 Report
This manuscript introduced methods of summary measures of health inequality, including their definition, calculation, interpretation and application. It is valuable in theory, methodology and practice.
Considering that this paper is more focused on the application of the HEAT package and not much discussion on the history, status, problems and trends of health equity assessment methods, I would suggest adding the words "in the WHO's Health Equity Assessment Toolkit" or "in the HEAT" in the title.
Author Response
Dear reviewer,
Thanks very much for your favorable review. Please see the attachment as our response to your point.
Best regards,

Reviewer 2 Report
Summary
The manuscript provides an introduction to the various health inequality measures contained in the WHO Health Equity Assessment Toolkit (HEAT). The review focuses on how the measures are calculated, with an application to three indicators in Indonesia over the period 1997 to 2017. The manuscript would be useful for researchers and analysts working with the HEAT tools, but reads more like a technical report than a research article.
Detailed points:
* The title is misleading, since it suggests a broader review than just those measures contained within HEAT. This should be made more specific, or the scope of the paper should be broadened (in terms of discussion) to cover measures that are not part of HEAT (e.g. the Gini coefficient, which despite its weaknesses is still in use).
* It would be interesting to learn more about how uncertainty estimates are implemented for the different measures, and to see the uncertainties reflected in the Indonesian application example. At present, there is just a single sentence (line 137-138) dedicated to this important topic.
* It would be interesting for the authors to highlight and discuss the interpretation of linear versus nonlinear measures. For difference and ratio measures, the authors make similar interpretations, i.e. the further from zero (or one for ratio), the higher the inequality. However the 'furtherness' cannot be interpreted in the same way in both cases. If the ratio measure were converted to a difference measure by applying a log transformation, would the interpretation be different?
* The Indonesian example provides a clean illustration of how all indicators can be telling the same story. Are there any different examples where indicators are not doing this, and do these provide any insights into the situation in the jurisdiction?
* It may be an issue with the generation of the PDF document, but the schematics in Figure 1 and Figure 6 have missing/misformatted text. In Figure 6, the questions about weighting seem to be redundant for all options except for complex non-ordered measures. The answer to the question leading to simple measures should be "2" rather than "2 or >?"
Author Response
Dear reviewer,
Thanks very much for your favorable review. Please see the attachment as our responses to your points.
Best regards,
